# Phenolic Compounds and Organic Acid Composition of *Syringa vulgaris* L. Flowers and Infusions

**DOI:** 10.3390/molecules28135159

**Published:** 2023-07-01

**Authors:** Monika Gąsecka, Agnieszka Krzymińska-Bródka, Zuzanna Magdziak, Piotr Czuchaj, Joanna Bykowska

**Affiliations:** 1Department of Chemistry, Poznań University of Life Sciences, 60-637 Poznań, Poland; monika.gasecka@up.poznan.pl (M.G.); zuzanna.magdziak@up.poznan.pl (Z.M.); 2Department of Ornamental Plants, Dendrology and Pomology, Poznań University of Life Sciences, 60-637 Poznań, Poland; piotr.czuchaj@up.poznan.pl (P.C.); joanna.bykowska@up.poznan.pl (J.B.)

**Keywords:** phenolic compounds, organic acids, flowers infusion, *Syringa vulgaris* flowers

## Abstract

The study aimed to determine the content of phenolic compounds (phenolic acids and flavonoids) and organic acids in dried flowers and water infusions of non-oxidised and oxidised flowers from four lilac cultivars. The diversity in the total phenolic and flavonoid content was in the flowers (18.35–67.14 and 2.03–2.65 mg g^−1^ DW, respectively) and infusions (14.72–47.78 and 0.20–1.84 mg per 100 mL infusion, respectively) depending the flower colour and form (oxidised and non-oxidised). Phenolic compounds and organic acids were susceptible to oxidation. Compared to infusions, flowers had more phenolic compounds and organic acids. The highest content of most phenolic compounds was confirmed for non-oxidised purple flowers (up to 7825.9 µg g^−1^ DW for chlorogenic acid) while in infusions for non-oxidised white flowers (up to 667.1 µg per 100 mL infusions for vanillic acid). The phenolic profile of the infusions was less diverse than that of flowers. The scavenging ability ranged from 52 to 87%. The highest organic acid content in flowers was for oxidised blue and purple flowers (2528.1 and 2479.0 µg g^−1^ DW, respectively) while in infusions the highest organic acid content was for oxidised purple flowers (550.1 µg per 100 mL infusions).

## 1. Introduction

The growing demand for new nutraceutical plant foods has sparked interest in edible flowers. Edible flowers that grow in the human environment can be a source of many desirable chemicals of a health-promoting nature [1]. One of the long-lived deciduous shrubs with white, lilac, pink, and purple flowers is *Syringa vulgaris* (common lilac). In temperate climates, flowers bloom in April and May [2]. *Syringa vulgaris*, native to the Balkan peninsula [3], is cultivated as an ornamental species on the European continent [4] and in part in the United States [2]. Plants also make volatile oils, bactericides, and food additives [5]. Flowers are crystalized in salads with egg whites and sugar [6]. Flowers are used to prepare salads and infusions [7]. Yigit et al. [2] found that herbal tea is made from lilac flowers. Kalemba-Drożdż [7] suggested the preparation of the oxidised lilac tea because of the caramel taste. Flowers are withering, crushing, and heating at room temperature or in an oven at 40 °C. The process is similar to the production of black tea. Tanaka and Matsuo [8] maintain that the withered leaves are crushed in black tea production. After that, the polyphenols are oxidised. Enzymes are inactivated by heating and drying. The enzymatic oxidation in this process is sometimes called fermentation. It is not honest fermented tea because microorganisms do not participate. After harvesting, prepared leaves influence the tea’s taste and aroma quality [9].

*Syringa vulgaris* has anti-inflammatory, adaptogenic, and immunomodulatory properties [10]. Flowers are used in traditional medicine as antipyretic [11], appetizers [4], and to eliminate internal parasites [12]. In Greece, flower infusions are used internally to treat gastrointestinal problems and externally as a massage for gout and rheumatism [4]. Some flower chemical compounds can provide a new perspective for the clinical treatment of Parkinson’s disease [11]. Flower extracts represent valuable sources of compounds with antioxidant and antitumor potential [4].

The researchers focus mainly on the phenolic profile of the bark and leaves of lilac [10,13,14]. A limited number of papers can be found on the phytochemical composition of flowers [10,15].

This study aimed to determine the content of phenolic compounds (phenolic acids and flavonoids) and organic acids in flowers and water infusions of oxidised and non-oxidised *Syringa vulgaris* flowers.

## 2. Results

### 2.1. Phenolic Compounds

The total phenolic content (TPC) in the flowers ranged from 18.4 mg GAE g^−1^ for oxidised white flowers to 67.1 mg g^−1^ GAE for blue, non-oxidised flowers (Figure 1). The TPC was as follows for the non-oxidised flowers: blue ≥ pink > purple ≥ white. The oxidation resulted in a significant reduction in the TPC and was as follows: pink > blue ≥ purple > white. The infusion of TPC ranged from 14.7 for oxidised pink to 47.8 mg (for non-oxidised purple). The higher TPC was for the non-oxidised flowers compared to oxidised. The TPC for non-oxidised flowers was as follows: purple > pink ≥ white > blue. For oxidised flowers, it was as follows: blue > white ≥ purple > pink.

The total flavonoid content (TFC) in flowers ranged from 2.1 mg g^−^^1^ for oxidised pink flowers to 2.7 mg g^−^^1^ for non-oxidised blue flowers (Figure 2). The TFC of blue, pink, and white flowers was higher for the non-oxidised form while the TFC of purple flowers was similar for both states. The TFC in infusions ranges from 0.2 mg per 100 mL for oxidised blue flowers to 1.8 mg per 100 mL for non-oxidised pink flowers. The TFC was >1 mg in infusion in 6 variants (except oxidised white and blue flowers with a TFC ~ 0.2 mg). Furthermore, the TFC was similar for pink, white, and purple flowers (non-oxidised).

Phenolic compounds belonging to various subgroups have been confirmed in lilac flower (Table 1, Figure 3). Hydroxybenzoic acids with the C6C1 structure (gallic, 2,5-DHBA (2,5-dihydroxybenzoic acid), 4-HBA (4-hydroxybenzoic acid), vanillic and syringic acids), and hydroxycinnamic acids with the C6C3 structure (caffeic, p-coumaric, chlorogenic, ferulic, sinapic, and t-cinnamic acids) were quantified in all flowers. Protocatechuic acid (C6C1) was not determined in oxidised blue and white flowers. Among flavonoids, catechin, quercetin, and rutin were quantified in all flowers. However, apigenin was not detected in non-oxidised blue and luteolin in white and purple flowers.

The sum of detected phenolic compounds was very diverse. Among the non-oxidised flowers, purple flowers were the richest in the compounds studied, followed by blue, white, and pink flowers.

The sum of detected phenolic compounds was 4 to 19 times higher for non-oxidised flowers compared to oxidised forms.

Generally, the higher content of almost every single phenolic compound was confirmed in non-oxidised flowers. In addition to some exceptions, the oxidised purple flowers characterised the highest range of almost all phenolic compounds. However, the white, non-oxidised flowers contained the most elevated ferulic and trans-cinnamic acids and rutin content. Non-oxidised blue flowers characterise the most elevated chlorogenic and 2,5-DHBA, sinapic acid, and quercetin content.

Catechin and ferulic acid were dominant in white flowers without oxidisation while chlorogenic acid and catechin were confirmed as the main compounds in pink, blue, and purple flowers without oxidation. Generally, oxidation resulted in quantitative changes in the profile of phenolic compounds compared to non-oxidised flowers. Catechin was one of the main phenolic compounds in all oxidised flowers, next to protocatechuic acid (oxidised pink flowers) and chlorogenic acid. The reduction of the content of phenolic compounds was observed with some exceptions—for ferulic acid (oxidised blue and oxidised purple flowers) and 2,5-DHBA content (oxidised purple flowers).

The phenolic profile for the infusions was different from that for the flowers (Table 2). The richest in phenolic compounds was infusion from non-oxidised white flowers. Compared to flowers, the sum of quantified phenolic compounds was lower. Kaempferol was a new compound compared to flowers quantified in all infusions. The highest content of some compounds (p-coumaric, gallic, 4-HBA, t-cinnamic and vanillic acids, and kaempferol) was estimated for non-oxidised white flowers. In a profile of white flowers that were non-oxidised, the dominant was 4-HBA and vanillic acid (>600 µg g^−1^) while for oxidised flowers, 2,5-DHBA was the main compound (152 µg g^−1^). 4-HBA was the main phenolic in the infusions of pink flowers. The profile of the infusion of blue flowers was lower. The infusion of non-oxidised white flowers and syringic acid was dominant for non-oxidised flowers while 2,5-DHBA was for oxidised flowers. The phenolic profile for purple flower infusions was very diverse and rich in phenolic compounds with 2,5-DHBA as the main compounds for oxidised flowers and 4-HBA for oxidised.

### 2.2. Radical Scavenging Ability

The antioxidant activity was measured using the DPPH methods after 60 min of incubation and the ABTS method after 10 min of incubation. The results for the ethanolic extract show that the most remarkable ability to scavenge the DPPH radicals was confirmed for the non-oxidised flowers in addition to the blue flowers (Figure 4). Extracts of oxidised blue and non-oxidised pink flowers scavenged >80% of the radicals while the lowest ability was confirmed for non-oxidised blue and oxidised purple flowers (~66%). The scavenging effect towards DPPH radicals for infusions ranged from 68.7% for oxidised white flowers to 80.5% for oxidised blue flowers. Furthermore, a scavenging effect was confirmed only for oxidised blue flowers. Significant differences between infusions from oxidised and non-oxidised flowers were only for blue flowers.

The results of the ABTS method pointed to the significantly higher RSA towards ABTS cation radicals for all extracts from non-oxidised flowers with a similar value for pink, blue, and white flowers (>72%) and lower RSA for purple flowers (67.5%, Figure 5). RSA for non-oxidised flowers was between 51.9 (for purple flowers) to 62.0% (for white flowers). The RSA for infusion was between 55.2 (non-oxidised blue) to 78.1 % (oxidised blue flowers). Oxidation resulted in a drop in the RSA for purple and white flowers and elevation of the RSA for blue flowers.

### 2.3. Organic Acids

The profile and content of organic acids in the flowers of *S. vulgaris* depended on the colour variables tested and the method of preparation of the lilac flower samples (Table 3, Figure 3).

The highest acid content was found for blue and purple flowers. For both cultivars, a significant increase in the sum of determined acids was found in *S. vulgaris* flower samples after oxidation. The highest concentration of acetic, malonic, quinic, and succinic acid was determined in the non-oxidised blue flowers. After oxidation, a decrease in acetic and succinic acid was found while the content of malonic and quinic acids increased significantly. A high citric acid content was also determined in the flowers after oxidation, which was not found in the non-oxidised flowers. The highest concentrations of acetic, citric, malonic, oxalic, and quinic acids were found in purple flowers, and their amount increased significantly after oxidation.

The white and pink flowers were characterised by a significantly lower content of organic acids than the purple or blue flowers. Their content was significantly reduced after oxidation. In the white flowers, the highest content of acetic, malic, oxalic, quinic, and succinic acids was determined, which after oxidation, decreased significantly. Citric and malonic acid increased significantly after oxidation. The same relationships were shown for the pink flowers.

The organic acid content in water infusions of flowers was significantly lower than in flowers (Table 4). However, it was confirmed that the content of organic acids is the highest, both in the infusion and in the flowers, for purple flowers after the oxidation process. The highest malonic, malic, and lactic acid contents were determined in these samples. The presence of citric acid, present in flowers was not determined. For white and blue flowers, higher contents of organic acids were also found after oxidation.

However, these values were significantly lower compared to purple flowers. The dominant acid was succinic acid for the white flowers while the highest content of malonic and malic acids was found for the blue flowers. On the other hand, for the pink flowers, the highest acid content was found in non-oxidised flowers, and the dominant acids were succinic acid, followed by malonic acid, acetic acid, and lactic acid. The same relationship was found in white flowers, but the acetic acid content was the highest. The differences between flower extracts and flower infusions result from the difference in solvents and the extraction time.

## 3. Discussion

Infusions from different plants are becoming increasingly popular due to proven health-promoting properties. Common lilac is mainly valued for its ornamental quality. However, *Syringa vulgaris* flowers that are edible and rich in bioactive compounds (i.e., flavonoids, essential oils, iridoid glycosides, organic acids, and lignane glycosides) have been documented, which were also determined in other parts of the plant [2,3,4,5,13,14,16]. In connection with this, there has recently been a great interest in the therapeutic effect from antioxidants of *Syringa vulgaris* aromatic infusions containing health-promoting properties prepared mainly from flowers. Lilac flowers have been shown to contain the highest content of phenolic, total flavonoid, and total phenolic acid compared to other parts of the plant [4]. The research presented pointed to a diversity in the total content of phenolic and flavonoid dependent on the colour of the flowers in flowers and infusions. The TPC in the lilac flower and infusion analysed was lower (14–68 mg per 100 mL = mg GAE g^−1^) compared to black tea leaves (50–179 mg GAE g^−1^ dry leaf at 60–100 °C) [17]. In the most popular tea in the United Kingdom, the TPC was between 80–134 mg GAE g^−1^ of the DM for black tea and 87–106 mg GAE g^−1^ of the DM for green tea [18]. The highest TPC in the flowers was confirmed for blue and pink flowers. Oxidation resulted in a significant reduction in the TPC in flowers. The TPC was ~ 3-fold higher for non-oxidised blue flowers and ~ 2 for pink flowers that were not oxidised compared to oxidised flowers. It is worth underlining that the non-oxidised purple and the white flowers and oxidised pink flowers had a similar TPC. Thus, the oxidation process for pink flowers allowed one to obtain the TPC at the level of values for non-oxidised purple and white flowers. In infusions, the reduction in the TPC for oxidised forms, was also confirmed. The TPC was higher by nearly three times for purple and pink flowers and more than two times for white flowers.

The measurement of absorbance at 510 nm for a mixture and NaNO_2_, AlCl_3_, and NaOH extracts is one of the methods to determine the total flavonoid content. However, it should be remembered that the method is specific to catechins, rutin, luteolin, and phenolic acids [19]. Therefore, the results depend on the structure of the individual flavonoids present. For flowers and infusions, the lower TFC was for oxidised flowers. Furthermore, oxidation also reduced the total flavonoid content. No significant changes in the TFC were confirmed in infusions for oxidised purple, pink, and white flowers.

Infusions from different plants are popular throughout the world due to their health-promoting properties. For example, the TPC in Amazonian herb infusions was between ~12 to ~75 mg g^−1^ GAE, and the TFC was between ~1 up to ~16 mg catechin equivalent g^−1^ DW [20].

Differences in the phenolic profile were visible in the colour of the flowers and the oxidised and non-oxidised forms. Purple and blue flowers were very rich in phenolic compounds. The previous study confirmed that the TPC, TFC, phenolic profile, and antioxidant properties in different herbal teas depend on the effect of the flower drying technique, method, and brewing time [2,21,22]. The drying method is one of the key parameters that influences the content of phenolic compounds in lilac flowers and their subsequent use [2]. The oven drying resulted in the level of a decrease in the TPC and scavenging activity. However, the content of individual components can increase and decrease [2]. The flowers are used to prepare infusions.

The phenolic profile of flowers was varied, especially for the non-oxidised flowers. The content was dependent on the colour of the flowers. Catechin (620.1–3019 µg g^−1^) was one of the dominant phenolic compounds for all flowers that were non-oxidised. The highest content of individual phenolic compounds for chlorogenic acid (7825.9 µg g^−1^) was documented for non-oxidised purple flowers.

The research shows that all analysed phenolic compounds were susceptible to oxidation. The content of most phenolic compounds in the white flowers (in addition to flowers (in addition to catechin) and pink (besides catechin and protocatechuic acids) was very low (≤20 µg g^−1^). However, there was a spectacular drop in chlorogenic acid in oxidised purple flowers. It should be noted that some phenolic compounds, due to oxidation, showed different changes depending on the colour of the flowers, so the content of ferulic and gallic acids increased in blue flowers, 2,5-DHBA increased in purple flowers, and sinapic acid was stable in white flowers. Moreover, apigenin was confirmed to be a new compound in oxidised blue flowers compared to non-oxidised ones.

The phenolic profile of the infusion was less diverse than that of the flowers. However, a very high content (>600 µg g^−1^) was estimated for vanillic acid oxidised (white flowers without oxidisation) and 2,5-DHBA (non-oxidised purple flower). Not every infusion of oxidised flowers confirmed a decrease in the content of phenolic compounds. Many studies indicated that extracts and infusions of different herbs and plants have many phenolic compounds belonging to other groups and derivatives from phenolic acids [23,24]. Furthermore, the infusion time, the solvent used for the extraction, and the material harvest time affected the quantitative and qualitative composition [25,26,27].

The scavenging capacity of ethanolic extracts toward DPPH radicals for all was between 66–87%, with a higher value for non-oxidised flowers compared to non-oxidised while for infusions, the higher scavenging ability was for oxidised pink and blues flowers compared to non-oxidised. Furthermore, the scavenging capacity of ethanolic extracts was correlated with the TPC (0.925), TFC (0.713), 2,5-DHBA (0.554), protocatechuic acid (0.413), and sinapic acid (0.499), but there was no correlation between the scavenging capacity and TPC and the TFC for infusion. The scavenging ability toward ABTS cation radicals was between 52 to 78% for all analysed samples. The significant correlation of the RSA of the flower extract (toward the ABTS radicals) was with the TPC (0.801), TFC (0.606), caffeic acid (0.646), chlorogenic acid (0.464), rutin (0.579), quercetin (0.420), t-cinnamic acid (0.672), and sum of phenolic compounds (0.438) while for infusions only for chlorogenic acid (0.533) and protocatechuic acid (0.474). Previous studies documented that the antioxidant activity of lilac extract depends on drying methods, type of tissue, type of solvent used for extraction, and extraction and time of extraction [2,10,15].

In terms of phenolic compounds, the significant differences in the content and profile of organic acids were also found. Their level was influenced by the colour of the *S. vulgaris* flowers and their preparation. Both factors determined the total content of the determined acids, but above all, they impacted the content level and profile. Organic acids are important metabolites in the carboxylic acid cycle; therefore, the analysis of their content is important due to their participation in various physiological processes of the cell, such as respiration and energy production, amino acid biosynthesis, photosynthesis, cation transport, and reduction of stress caused by the presence of heavy metals [28,29,30], but also their taste and nutritional properties [31]. The highest contents of acetic, citric, malic, oxalic, quinic, citric, and succinic acids were found in the flowers of *S. vulgaris*. They can increase the acidity of the infusion in the case of citric acid [32,33], providing a pleasant taste when malic acid is present [34]. It is also indicated that succinic acid and sodium salt are among the most intense substances that contribute to a particularly sour taste [35]. Malic acid improves muscle performance, reduces fatigue, and improves mental clarity [36]. Its high content was found in the leaves of *Rumex dentatus* L., *Cannabis sativa* L. [37], and the petals of *Rosa* and *Calendula officinalis* L. [31]. In lilac flowers, the highest content of malic acid was found in purple flowers, and the content was almost identical in non-oxidised and oxidised flowers and white flowers only after the process that was non-oxidised. Fumaric acid was determined at lower concentration levels, as in the work of Pires et al. [31]; however, its highest content was found to be pink and purple in non-oxidised flowers. Its presence is important due to its attributes (e.g., antioxidant, anti-inflammatory, analgesic, chemo preventive, anti-psoriasis, immunomodulatory, and neuroprotective properties) [38]. Both citric and malic acids are added to food as natural preservatives and prevent blackening [32,34].

Infusions made from lilac flowers were characterised by a significantly lower content of organic acids in flowers. Other studies also confirm where infusions are poorer in the compounds [31]. However, compared to the petals and infusions of other flowers, it can be clearly stated that the content of organic acids in *S. vulgaris* is significantly lower, regardless of the colour and method of preparation. In the work of Pires et al. [31], among infusions, the highest concentrations were found in *Calendula* infusions, mainly due to the presence of quinic and succinic acids (14.5 and 11.2 mg per 100 mL of infusion, respectively) and in the *Centaurea cyanus*, which had the highest contents of quinic and citric acid (7.4 and 15.5 mg 100 mL^−1^ of infusion, respectively) [31]. In turn, in *Ballota nigra*, the high content of organic acids (approximately 14 g kg^−1^) was identified, with quinic acid as the main compound, which corresponds to 58.9% of the sum of the determined acids [39]. After oxidation, the infusion’s highest content of 550 µg 100 mL^−1^ of organic acids was determined for the blue flowers. It should be emphasised that the flowers were at least 4.5 times richer in organic acids for the blue colour and even up to 360 times richer for the non-oxidised blue flowers. The content of organic acids in elderberry extracts was shown to depend on the drying method, the type of tissue, the type of solvent used for extraction, and the extraction time.

## 4. Materials and Methods

### 4.1. Characteristics of Experimental Materials and Their Preparation

Four lilac cultivars were analysed (Figure 6): Liliana (white flowers), Prof. Hoser (light blue flowers), Jules Simon (light pink flowers), and Andenken an Ludwig Späth (purple flowers). The flowers were obtained from the 60-year-old collection of the Institute of Dendrology of the Polish Academy of Sciences in Kórnik, Western Poland. The flowers were harvested in May 2021 and divided into two batches. The first batch was dried for 5 days in darkness at room temperature at 20 °C and humidity 40%. The second batch was prepared: withering for 12 h, mechanically crushed (rolling), strictly putting and closing in glass jars for 24 h at 25 °C, and drying for 5 days in darkness at 20 °C and humidity 40%. The flowers were stored at 20 °C in closed glass jars in darkness and humidity 3%, and in April 2023, the phenolic compounds and the organic acid content were analysed.

### 4.2. Experiment Design, Sample Preparation

The flower samples were homogenized in liquid nitrogen. Then, 80% ethanol was added. The samples were sonicated for 20 min at 30 °C (Bandelin Sonorex RK 100, Berlin, Germany) and then shaken for 12 h at room temperature (Ika KS 260 shaker, IKA-Werke GmbH & Co. Kg, Staufen, Germany). The samples were centrifuged at 3600 rpm/min for 15 min at 25 °C (Universal 320R Hettich Zentrifugen, Tuttlingen, Germany). The samples were dried and, before analyses, redisolved in 80% ethanol.

The 1 g of flowers was poured with hot water at 95 °C and left under cover for 10 min. The infusion was filtered. After being cooled, the infusion was concentrated in the evaporator. The extraction procedure is presented at Figure 7.

### 4.3. Determination of Phenolic Compounds and Organic Acids

The organic acid and phenolic compounds of flowers and infusions were quantified according to Gąsecka et al. [40] and Magdziak et al. [30] (using ultra-performance liquid chromatography (ACQUITY UPLC H-Class System Waters Corporation, Milford, MA, USA) consisting of a quaternary pump solvent management system, an online degasser, and an auto sampler. Acquity UPLC BEH C18 column (2.1 mm 9 150 mm, 1.7 lm, Waters), thermostated at 35 °C, was used for all analyses. The flow rate was 0.4 mL min^−1^ for the gradient elution with water and acetonitrile (both containing 0.1% formic acid, pH = 2). The gradient programme was as follows: flow 0.4 mL min^−1^ −5% B (2 min), 5–16% B (5 min), 16% B (3 min), 16–20% B (7 min), 20–28% B (11 min) flow 0.45 mL min^−1^ −28% (1 min), 28–60% B (3 min) flow 5.0 mL min^−1^ −60–95% B (1 min), 65% B (1 min), 95–5% B (0.1 min) flow 0.4 mL min^−1^ 5% B (1.9 min). The identification of peaks was based on a comparison with the retention times of chemical standards. Using an external standard, the detection was performed in a Waters Photodiode Array Detector (Waters Corporation, Milford, MA, USA) at λ = 280 nm (catechin, gallic acid, 4-hydroxybenzoic acid, protocatechuic acid, syringic acid, t-cinnamic acid, vanillic acid, and organic acids: acetic, citric, fumaric, lactic, malic, malonic, oxalix, quinic, suscinic) and λ = 320 nm (apigenin, caffeic acid, chlorogenic acid, *p*-coumaric acid, ferulic acid, kaempferol, luteolin, rutin, quercetin, sinapic acid) (Table 5). Raw data were acquired and processed with Empower 3 software. Before injection, the extracts were filtered through a 0.22 mm syringe filter.

### 4.4. Determination of the Total Phenolic Content

Total phenolic content (TPC) was determined with Folin–Ciocalteu reagent according to Singleton and Rossi [41] with some modifications. The 100 µL of ethanolic flower extract/infusion was mixed with 1 mL of Folin–Ciocalteu phenol reagent, and after 3 min, 3 mL of 10% Na_2_CO_2_ was added. The samples were kept in the dark for 30 min at room temperature. The absorbance at λ = 765 nm was then measured with a UV spectrophotometer (Carry 300 Bio UV-visible spectrophotometer (Varian, Agilent, Santa Clara, CA, USA)). Results were expressed in mg of gallic acid equivalents per g of dry weight (mg GAE g^−1^) and mg of gallic acid equivalents per 100 mL of infusion. The calibration curve range was 0.01 to 0.5 mg mL^−1^ (R^2^ = 0.981).

### 4.5. Determination of the Total Flavonoid Content

The total flavonoid content was measured according to Zhuang et al. [42] with some modifications: 200 µL of ethanolic extract/infusion and 75 µL of 5% NaNO_2_ were mixed. After 6 min of incubation, 150 µL of 10% AlCl_3_ was added. After the next 6 min, 4 mL of 1M NaOH was added. The absorbance was measured at 510 nm measured with a UV spectrophotometer (Carry 300 Bio UV-VisibleSpectrophotometer (Varian, Agilent, Santa Clara, CA, USA), and TFC was expressed as mg of quercetin equivalents per g of dry weight and mg of gallic acid equivalents per 100 mL of infusion). The calibration curve range was 0.01–0,5 mg^.^mL^−1^ (R^2^ = 0.975).

### 4.6. DPPH Radical Scavenging Assay

Inhibition of the 2,2- diphenylhydrazyl (DPPH) radical was measured according to Stojichevich [43] with some modifications. To determine the radical scavenging ability, 1 mL of ethanolic extract of 10 mg^.^mL^−1^ was mixed with 2.7 mL of 6 µmol^.^L^−1^ methanolic solution of 2,2-diphenyl-1-picrylhydrazyl (DPPH) radicals. The shaken mixtures were kept in the dark at room temperature for 60 min. The absorbance was measured at 517 nm with a UV spectrophotometer (Carry 300 Bio UV-VisibleSpectrophotometer, Varian, Agilent, Santa Clara, CA, USA). The DPPH radical scavenging activity was calculated as the reduction of the DPPH radical according to a formula [1].

### 4.7. ABTS Radical Scavenging Assay

To generate ABTS radical cations 7 mol^.^L^−1^ of ABTS (2,20-azino-bis-(3-ethylbenzothiazoline-6-sulfonic acid) diammonium salt) to 2.45 mol l^−1^ potassium persulfate solution were mixed and incubated overnight in the dark at room temperature. The ABTS solution was diluted with distilled water to obtain an absorbance of 1.4–1.5 at 734 nm [44]. The ABTS solution was mixed with ethanolic extract, and after 10 min, the absorbance at 734 nm was measured with a UV spectrophotometer (Carry 300 Bio UV-VisibleSpectrophotometer, Varian, Agilent, Santa Clara, CA, USA). The radical scavenging activity (RSA) was calculated as the percentage of ABTS discoloration according a formula [1]. The DPPH and ABTS^+^ radical scavenging activity was calculated according to the formula [1]:RSA (%) = (A − AC)/A × 100
where A was absorbance of the control (DPPH or ABTS solution without extract) and AC was absorbance of ethanolic extract.

### 4.8. Chemicals

The standards of phenolic compounds (gallic acid ≥98%, protocatechuic acid ≥ 98.99%, 4-hydroxybenzoic acid ≥ 99%, 2,5–dihydroxybenzoic acid ≥98%, vanillic acid ≥ 97%, syringic acid ≥ 98%, catechin ≥ 98%, caffeic acid (certified reference material TraceCERT^®^), p-coumaric acid ≥ 98%, ferulic acid ≥ 98%, chlorogenic acid ≥ 95%, sinapic acid ≥ 97%, rutin ≥ 94%, trans-cinnamic acid ≥ 99%, quercetin ≥ 98%, luteolin ≥ 98%, naringenin ≥ 95%, apigenin ≥ 95%, kaempferol ≥ 98%, vitexin ≥ 95%) and organic acids (acetic ≥ 99.7%, citric ≥ 99.5–100.5%, fumaric ≥ 99 %, lactic ≥ 85%, malic ≥ 99 %, malonic (certified reference material TraceCERT^®^), oxalic ≥ 99 %, quinic (analytical standard), and succinic ≥ 99.5 %) were purchased in Sigma–Aldrich (Steinheim, Germany) except for caffeic acid, which was obtained from Switzerland. Ethanol (absolute for EMSURE^®^ ACS, ISO, Reag. Ph Eur, ≥99.9 %) and 2.2-diphenyl-1-picrylhydrazyl (DPPH) (≥99%), 2,20 -azino-bis-(3-ethylbenzothiazoline-6-sulfonic acid) diammonium salt (≥98%), potassium persulfate) (≥99%) were purchased in Sigma–Aldrich (Saint Louis, MO, USA). Sodium nitrate (III) (sodium nitrite, NaNO_2_, pure p.a.), aluminium chloride (AlCl_3_, ≥ 99%), sodium hydroxide (NaOH, 100%) were purchased in Avantor Performance Materials Poland S.A. Pureliquid nitrogen was obtained from Air Products Sp. z o.o. (Warsaw, Poland).

### 4.9. Statistical Analysis

The results were expressed as mean values from tree replications. Data were processed using Microsoft Excel 2010. Statistical analysis was performed using STATISTICA 13.3 statistical software (StatSoft, Tulsa, Ok, USA) with two-way ANOVA followed by Tukey’s post hoc test (the results marked with identical superscripts in rows do not show differences at the significance level α = 0.05. Pearson’s correlation coefficients were estimated for pairs of parameters.

## 5. Conclusions

The results indicate that the flowers of *S. vulgaris* are a source of phenolic compounds and organic acids. The scavenging capacity was correlated with the total content of phenolic and flavonoid content in flowers. Quantitative and qualitative differences between infusions and extracts of oxidised and non-oxidised flowers showed that potential health benefits depend on how the flowers are processed for consumption.

The results indicate that the content of the bioactive compounds depended on the colour of the flower and its oxidation. Infusions prepared from these flowers are a valuable source of these bioactive compounds despite their varied content. Depending on the preferences of consumers’ infusions (different tastes and aromas depending on the colour and oxidation), their introduction to consumption may diversify and enrich the diet with the analysed bioactive compounds.

The novelty in this paper is the total phenolic content of the infusions of *S. vulgaris* flowers. In addition, in our work, oxidised lilac flowers were subjected to chemical analysis for the first time. In the next stage of research, other groups of bioactive compounds will be determined, including anthocyanins.

## Figures and Tables

**Figure 1 molecules-28-05159-f001:**
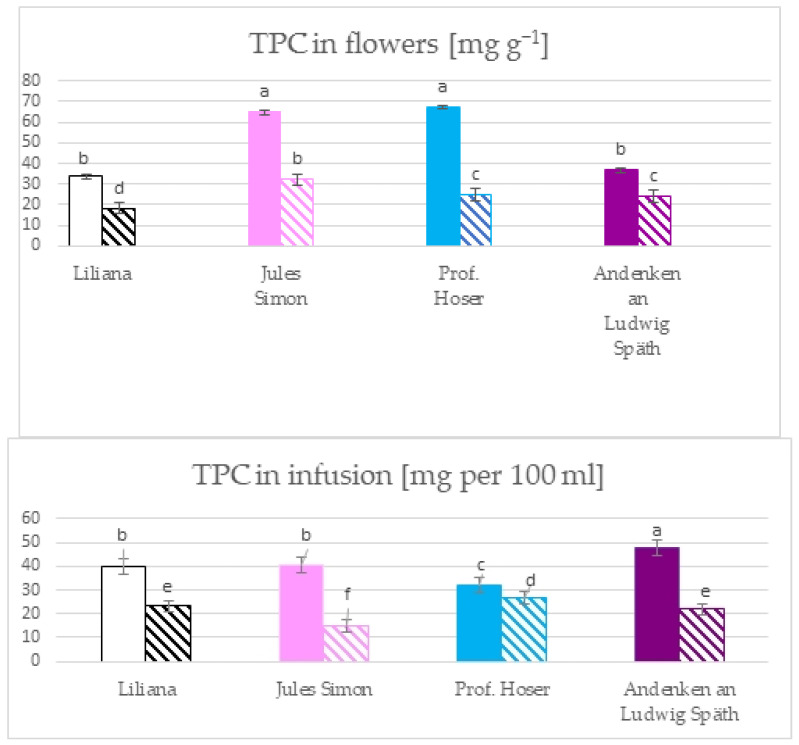
The total phenolic content in the flowers and the infusion of flowers from *S. vulgaris* flowers (*n* = 3; identical superscripts (a–f) denote non-significant differences between means according to the post hoc Tukey’s HSD test; non-marked area—non-oxidised flowers, marked area—oxidised flowers).

**Figure 2 molecules-28-05159-f002:**
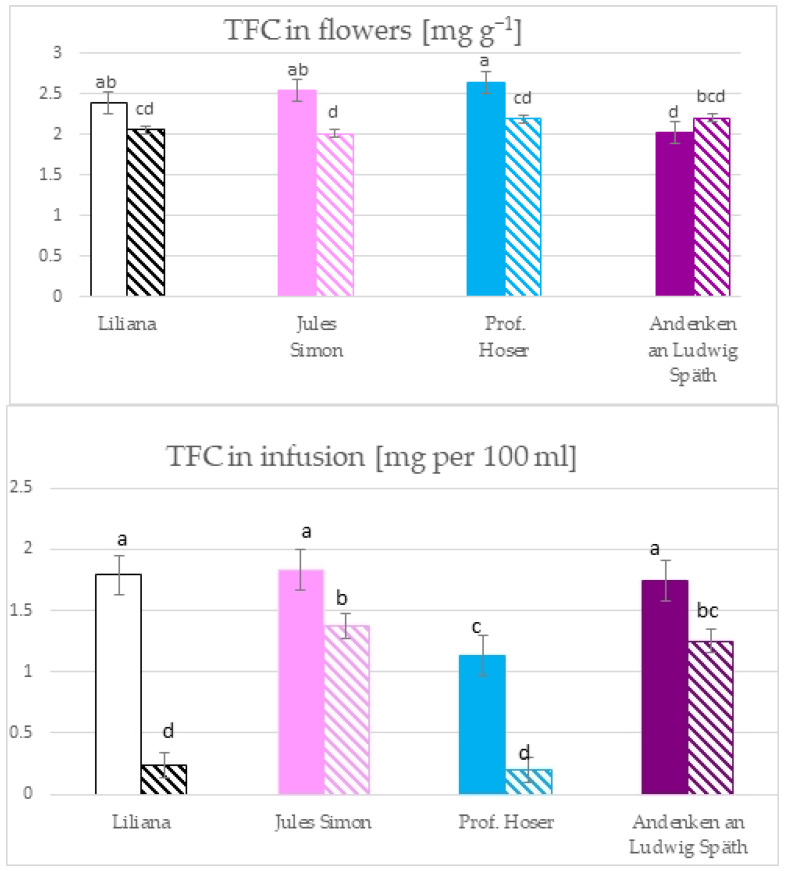
The total flavonoid content in the flowers and the infusion of *S. vulgaris* flowers (*n* = 3; identical superscripts (a–d) denote nonsignificant differences between means according to the post hoc Tukey’s HSD test; non marked area—non-oxidised flowers, marked area—oxidised flowers).

**Figure 3 molecules-28-05159-f003:**
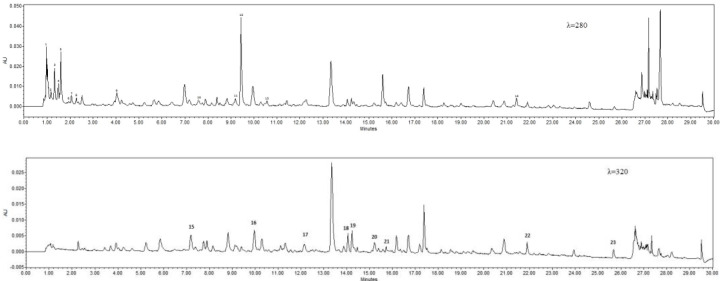
Sample chromatograms of oxidised flowers (λ = 280; oxalic acid, 2. quinic acid, 3. malonic acid, 4. lactic acid, 5. citric acid, 6. succinic acid, 7. fumaric acid, 8. gallic acid, 9. protocatechuic acid, 10. 4-hydroxybenzoic acid, 11. vanillic acid, 12. catechin, 13. syringic acid, 14. t-cinnamic acid; λ = 320, 15. 2,5-dihydroxybenzoic acid, 16. caffeic acid, 17. p-coumaric acid, 18. chlorogenic acid, 19. ferulic acid, 20. sinapic acid, 21. rutin, 22. quercetin, 23. apigenin.

**Figure 4 molecules-28-05159-f004:**
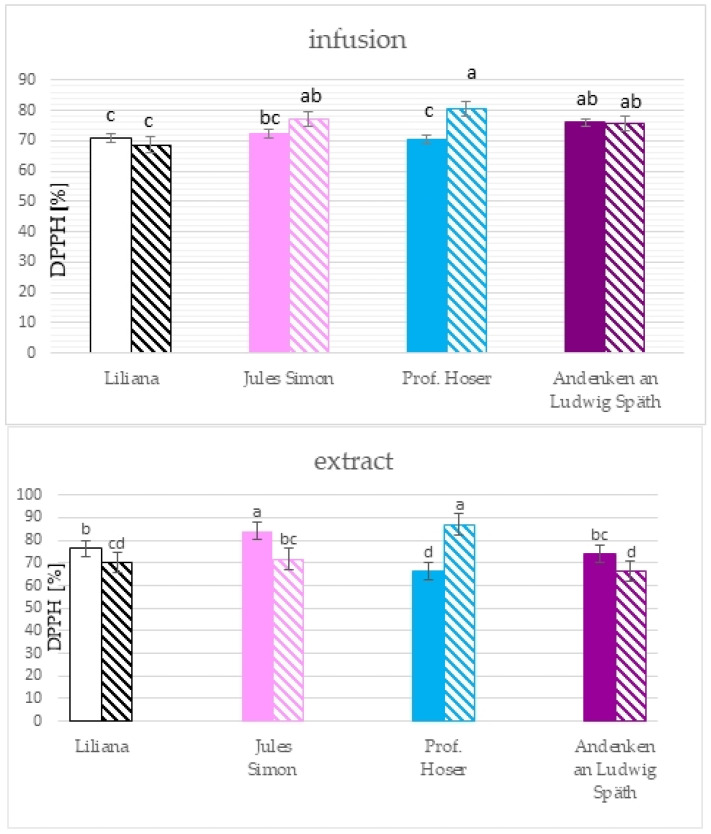
Scavenging activity of ethanolic extracts (10 mg^.^mL^−1^) and infusions of *S. vulgaris* flowers (*n* = 3; identical superscripts (a–d) denote non-significant differences between means according to the post hoc Tukey’s HSD test; non-marked area—non-oxidised flowers, marked area—oxidised flowers).

**Figure 5 molecules-28-05159-f005:**
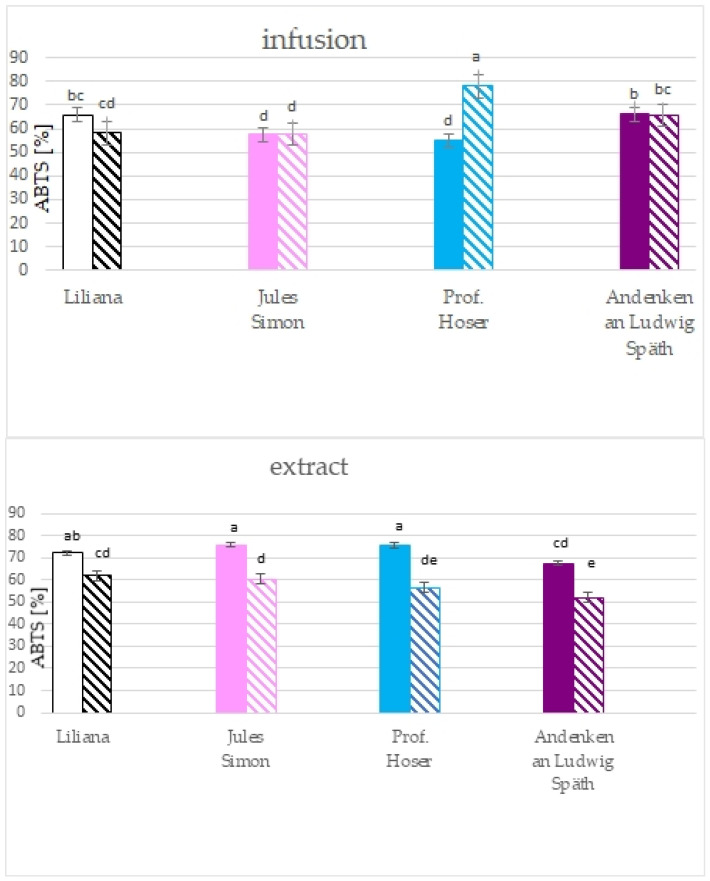
Scavenging activity of ethanolic extracts (10 mg^.^mL^−1^) and infusions of *S. vulgaris* flowers (*n* = 3; identical superscripts (a–e) denote non-significant differences between means according to the post hoc Tukey’s HSD test; non-marked area—non-oxidised flowers, marked area—oxidised flowers).

**Figure 6 molecules-28-05159-f006:**
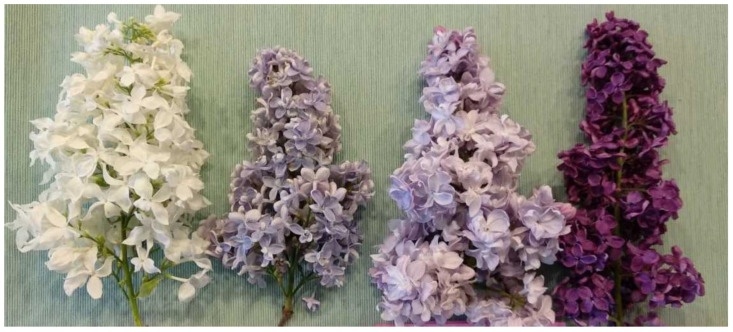
Cultivars of *S. vulgaris* (from left Liliana, Prof. Hoser, Jules Simon, and Andenken an Ludwig Späth).

**Figure 7 molecules-28-05159-f007:**
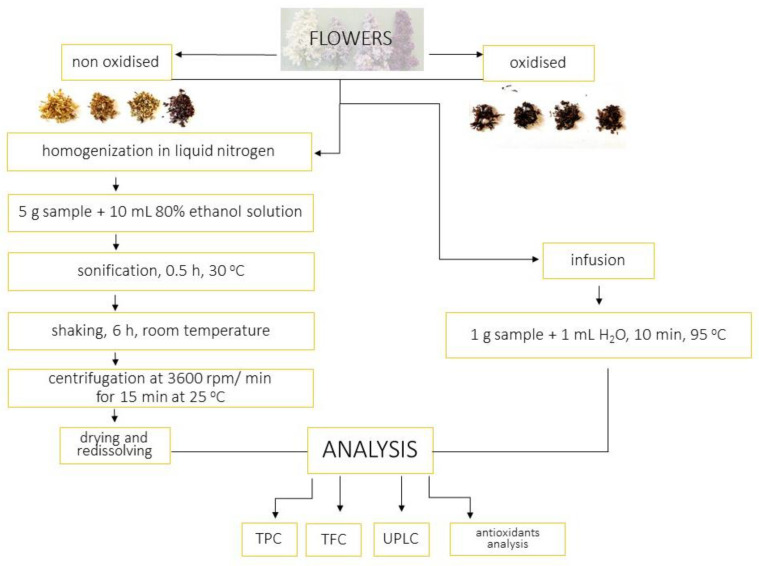
Extraction procedure.

**Table 1 molecules-28-05159-t001:** The content of phenolic compounds [µg g^−1^] in *S. vulgaris* flowers.

	*Syringa vulgaris*							
Compound	Liliana White Flowers		Jules Simon Pink Flowers		Prof. Hoser Blue Flowers		Andenken an Ludwig Späth Purple Flowers	
	Non-Oxidised	Oxidised	Non-Oxidised	Oxidised	Non-Oxidised	Oxidised	Non-Oxidised	Oxidised
Caffeic acid	755.7 ± 29.51 ^b^	19.0 ± 2.14 ^f^	335.4 ± 17.62 ^c^	9.9 ± 0.98 ^f^	653.4 ± 45.24 ^b^	114.9 ± 13.94 ^e^	793.8 ± 38.97 ^a^	189.7 ± 12.42 ^d^
Chlorogenic acid	492.4 ± 15.19 ^d^	15.9 ± 0.29 ^e^	1141.3 ± 56.34 ^c^	17.6 ± 1.55 ^e^	6114.6 ± 281.83 ^b^	159.6 ± 11.80 ^de^	7825.9 ± 334.93 ^a^	63.5 ± 3.46 ^de^
p-Coumaric acid	34.9 ± 24.7 ^b^	1.04 ± 0.05 ^e^	21.6 ± 1.12 ^c^	1.0 ± 0.06 ^e^	22.9 ± 1.35 ^c^	13.7 ± 3.09 ^d^	101.0 ± 5.42 ^a^	12.6 ± 0.79 ^e^
2,5-DHBA	224.6 ± 14.87 ^b^	12.5 ± 2.08 ^d^	19.9 ± 1.18 ^d^	5.3 ± 0.24 ^d^	879.4 ± 17.39 ^a^	86.8 ± 3.53 ^c^	11.6 ± 1.60 ^d^	220.6 ± 14.67 ^b^
Ferulic acid	944.2 ± 47.56 ^a^	2.1 ± 0.05 ^b^	6.5 ± 0.11 ^b^	1.1 ± 0.07 ^b^	7.0 ± 0.09 ^b^	30.6 ± 1.14 ^b^	8.3 ± 0.49 ^b^	28.3 ± 1.06 ^b^
Gallic acid	24.4 ± 2.80 ^d^	12.7 ± 1.38 ^d^	23.1 ± 1.52 ^d^	8.3 ± 0.10 ^d^	166.2 ± 14.06 ^b^	163.4 ± 8.51 ^b^	245.6 ± 10.35 ^a^	67.45 ± 2.27 ^c^
4-HBA	582.1 ± 20.32 ^b^	12.8 ± 2.28 ^e^	225.5 ± 13.23 ^d^	3.5 ± 0.05 ^e^	459.7 ± 16.89 ^c^	147.6 ± 12.80 ^e^	1763.8 ± 70.40 ^a^	126.7 ± 5.91 ^e^
Protocatechuic acid	nd	nd	165.9 ± 7.90 ^c^	47.7 ± 79.82 ^d^	319.3 ± 13.27 ^b^	nd	767.7 ± 33.82 ^a^	97.8 ± 4.15 ^cd^
Sinapic acid	12.6 ± 1.10 ^e^	10.3 ± 0.60 ^e^	75.9 ± 4.00 ^bc^	3.4 ± 0.16 ^e^	104.6 ± 7.23 ^a^	34.2 ± 1.78 ^d^	84.5 ± 4.96 ^b^	68.3 ± 2.34 ^c^
Syringic acid	89.2 ± 1.35 ^c^	7.9 ± 0.19 ^e^	45.4 ± 2.07 ^d^	3.8 ± 0.20 ^e^	62.7 ± 2.62 ^d^	153.1 ± 9.16 ^b^	382.4 ± 15.81 ^a^	49.7 ± 1.21 ^d^
t-Cinnamic acid	182.9 ± 9.87 ^a^	1.6 ± 0.11 ^e^	58.9 ± 1.21 ^c^	1.19 ± 0.17 ^e^	88.6 ± 1.73 ^b^	20.4 ± 1.23 ^d^	53.5 ± 1.65 ^c^	5.6 ± 0.30 ^e^
Vanillic acid	102.3 ± 5.69 ^d^	19.4 ± 0.77 ^e^	43.5 ± 3.95 ^e^	10.3 ± 1.17 ^e^	224.1 ± 20.68 ^b^	218.7 ± 18.09 ^b^	311.0 ± 16.09 ^a^	172.7 ± 11.54 ^c^
Apigenin	99.9 ± 7.37 ^b^	3.6 ± 0.16 ^e^	61.4 ± 2.33 ^c^	2.4 ± 0.10 e	nd	67.7 ± 2.10 ^c^	266.2 ± 11.01 ^a^	31.8 ± 1.63 ^d^
Catechin	1175.4 ± 42.04 ^d^	131.5 ± 18.25 ^e^	620.1 ± 23.35 ^d^	55.1 ± 4.81 ^e^	1684.4 ± 71.92 ^b^	1142.1 ± 33.51 ^c^	3019.0 ± 76.74 ^a^	1530.9 ± 60.51 ^c^
Luteolin	3.3 ± 0.10 ^c^	1.5 ± 0.06 ^d^	nd	nd	nd	nd	20.5 ± 1.01 ^a^	10.2 ± 0.72 ^b^
Quercetin	114.0 ± 9.83 ^b^	3.8 ± 0.08 ^e^	43.8 ± 1.46 ^d^	1.1 ± 0.07 ^e^	269.5 ± 4.13 ^a^	83.6 ± 5.40 ^c^	282.2 ± 16.15 ^a^	74.2 ± 3.78 ^c^
Rutin	217.6 ± 10.48 ^a^	17.4 ± 0.83 ^e^	158.5 ± 10.00 ^b^	2.8 ± 0.06 ^e^	115.3 ± 11.55 ^c^	94.5 ± 5.03 ^c^	227.9 ± 14.79 ^a^	65.2 ± 1.65 ^d^
Sum	5055.3 ± 133.92 ^c^	273.1 ± 14.84 ^e^	3046.9 ± 56.91 ^d^	174.4 ± 81.58 ^e^	11,172.0 ± 450.21 ^b^	2531.3 ± 10.81 ^d^	16,164.8 ± 612.45 ^a^	2815.6 ± 36.27 ^d^

*n* = 3; identical superscripts (a–f) denote nonsignificant differences between means in rows according to the post hoc Tukey’s HSD test; nd—not detected.

**Table 2 molecules-28-05159-t002:** The content of phenolic compounds [µg per 100 mL] of *S. vulgaris* flower infusions.

	*Syringa vulgaris*							
Compound	Liliana White Flowers		Jules Simon Pink Flowers		Prof. Hoser Blue Flowers		Andenken an Ludwig Späth Purple Flowers	
	Non-Oxidised	Oxidised	Non-Oxidised	Oxidised	Non-Oxidised	Oxidised	Non-Oxidised	Oxidised
Caffeic acid	62.5 ± 1.93 ^c^	1.7 ± 0.38 ^f^	146.1 ± 4.29 ^a^	53.6 ± 1.51 ^d^	nd	nd	73.8 ± 3.97 ^b^	14.8 ± 0.77 ^e^
Chlorogenic acid	52.2 ± 1.43 ^b^	2.5 ± 0.26 ^d^	21.6 ± 1.00 ^c^	1.9 ± 0.07 ^d^	nd	49.9 ± 1.27 ^b^	2.0 ± 0.43 ^d^	95.1 ± 10.09 ^a^
p-Coumaric acid	9.2 ± 0.45 ^a^	2.5 ± 0.30 ^d^	8.3 ± 0.42 ^b^	3.2 ± 0.20 ^cd^	nd	nd	3.5 ± 0.17 ^c^	nd
2,5-DHBA	312.1 ± 8.34 ^b^	152.6 ± 3.67 ^c^	102.4 ± 2.39 ^de^	68.7 ± 2.24 ^f^	nd	126.2 ± 5.45 ^cd^	653.7 ± 22.62 ^a^	78.3 ± 4.28 ^ef^
Ferulic acid	nd	nd	38.4 ± 1.42 ^a^	nd	1.8 ± 0.33 ^c^	nd	21.6 ± 1.41 ^b^	nd
Gallic acid	97.1 ± 2.96 ^a^	nd	nd	nd	nd	nd	7.3 ± 0.66 ^c^	34.3 ± 1.44 ^b^
4-HBA	627.5 ± 20.60 ^a^	16.1 ± 0.78 ^d^	250.3 ± 6.34 ^b^	201.2 ± 5.64 ^c^	nd	19.8 ± 0.39 ^d^	12.5 ± 0.98 ^d^	212.4 ± 2.62 ^c^
Protocatechuic acid	nd	nd	nd	nd	nd	60.1 ± 1.14 ^b^	7.1 ± 0.40 ^c^	135.0 ± 3.80 ^a^
Sinapic acid	7.9 ± 0.39 ^b^	54.8 ± 2.56 ^a^	3.3 ± 0.24 ^c^	nd	nd	nd	nd	nd
Syringic acid	191.8 ± 5.10 ^b^	26.9 ± 1.99 ^ef^	56.7 ± 1.15 ^d^	38.8 ± 1.76 ^de^	433.9 ± 22.54 ^a^	8.5 ± 0.62 ^f^	14.7 ± 0.91 ^ef^	135.9 ± 9.75 ^c^
t- Cinnamic acid	22.5 ± 1.18 ^a^	2.6 ± 0.37 ^c^	nd	nd	nd	nd	11.9 ± 0.89 ^b^	13.1 ± 1.16 ^b^
Vanillic acid	667.1 ± 25.46 ^a^	43.9 ± 2.58 ^c^	nd	nd	nd	nd	7.4 ± 0.57 ^c^	122.2 ± 3.43 ^b^
Apigenin	nd	nd	9.7 ± 0.56 ^b^	2.8 ± 0.19 ^c^	nd	nd	23.2 ± 2.43 ^a^	nd
Catechin	77.0 ± 1.72 ^b^	nd	nd	nd	nd	14.7 ± 0.51 ^c^	13.5 ± 0.80 ^c^	195.5 ± 9.50 ^a^
Kaempferol	272.9 ± 7.90 ^a^	50.2 ± 1.45 ^e^	125.4 ± 5.18 ^b^	60.6 ± 2.36 ^de^	5.5 ± 0.18 ^f^	9.8 ± 0.21 ^f^	71.2 ± 2.56 ^d^	90.3 ± 1.39 ^c^
Quercetin	20.5 ± 1.22 ^b^	nd	nd	nd	nd	nd	64.5 ± 2.75 ^a^	nd
Rutin	nd	17.6 ± 0.47 ^d^	21.7 ± 0.53 ^c^	8.5 ± 0.33 ^e^	nd	nd	56.5 ± 2.89 ^a^	40.3 ± 1.20 ^b^
Sum	2420.9 ± 59.62 ^a^	371.6 ± 6.00 ^e^	790.7 ± 0.59 ^d^	439.2 ± 6.84 ^e^	441.2 ± 22.21 ^e^	289.0 ± 4.86 ^f^	1044.4 ± 21.84 ^c^	1167.1 ± 40.18 ^b^

*n* = 3; identical superscripts (a–f) denote nonsignificant differences between means in rows according to the post hoc Tukey’s HSD test; nd—not detected.

**Table 3 molecules-28-05159-t003:** The content of organic acids [µg g^−1^] in *S. vulgaris* flowers.

	*Syringa vulgaris*							
Acid	Liliana White Flowers		Jules Simon Pink Flowers		Prof. Hoser Blue Flowers		Andenken an Ludwig Späth Purple Flowers	
	Non-Oxidised	Oxidised	Non-Oxidised	Oxidised	Non-Oxidised	Oxidised	Non-Oxidised	Oxidised
Acetic	346.0 ± 4.45 ^d^	55.4 ± 0.91 ^e^	385.1 ± 5.84 ^d^	nd	1320.7 ± 93.07 ^a^	1005.6 ± 111.33 ^b^	737.1 ± 4.90 ^c^	922.2 ± 53.27 ^b^
Citric	10.7 ± 0.62 ^d^	22.7 ± 2.10 ^c^	7.2 ± 0.04 ^de^	31.6 ± 1.78 ^c^	nd	183.1 ± 5.41 ^b^	30.3 ± 3.24 ^c^	236.5 ± 5.83 ^a^
Fumaric	0.3 ± 0.02 ^de^	0.2 ± 0.01 ^e^	4.3 ± 0.12 ^a^	0.6 ± 0.03 ^d^	0.9 ± 0.08 ^c^	0.4 ± 0.08 ^de^	2.0 ± 0.24 ^b^	0.5 ± 0.03 ^de^
Lactic	15.0 ± 0.86 ^c^	nd	16.2 ± 0.22 ^c^	39.2 ± 3.18 ^b^	42.2 ± 1.13 ^b^	nd	149.1 ± 3.19 ^a^	nd
Malic	29.6 ± 0.77 ^b^	1.3 ± 0.24 ^d^	nd	nd	12.2 ± 1.54 ^c^	nd	44.6 ± 2.45 ^a^	44.6 ± 2.78 ^a^
Malonic	25.9 ± 0.49 ^e^	100.6 ± 1.54 ^d^	17.6 ± 0.50 ^e^	37.0 ± 3.44 ^e^	383.0 ± 5.59 ^c^	998.6 ± 37.35 ^b^	344.0 ± 5.79 ^c^	1047.0 ± 12.28 ^a^
Oxalic	9.3 ± 0.13 ^d^	5.0 ± 0.24 ^ef^	7.3 ± 0.19 ^de^	1.6 ± 0.22 ^f^	20.5 ± 1.17 ^c^	50.7 ± 3.07 ^b^	9.5 ± 0.53 ^d^	73.8 ± 1.56 ^a^
Quinic	37.0 ± 0.23 ^d^	5.3 ± 0.27 ^e^	32.4 ± 2.99 ^d^	1.6 ± 0.22 ^e^	121.7 ± 4.37 ^c^	199.7 ± 9.11 ^a^	35.3 ± 1.99 ^d^	154.5 ± 5.88 ^b^
Succinic	73.9 ± 1.13 ^d^	6.3 ± 0.13 ^f^	50.2 ± 1.81 ^e^	3.5 ± 2.10 ^f^	259.2 ± 5.99 ^a^	90.0 ± 2.24 ^c^	135.8 ± 1.53 ^b^	nd
Sum	547.7 ± 3.90 ^d^	196.8 ± 1.32 ^e^	520.1 ± 9.08 ^d^	115.1 ± 5.18 ^e^	2160.4 ± 93.07 ^b^	2528.1 ± 63.4 ^a^	1487.7 ± 6.95 ^c^	2479.0 ± 54.64 ^a^

*n* = 3; identical superscripts (a–f) denote non-significant differences between means in rows according to the post hoc Tukey’s HSD test; nd—not detected.

**Table 4 molecules-28-05159-t004:** The content of organic acids [µg per 100 mL] of *S. vulgaris* flower infusions.

*Syringa vulgaris*								
Acid	Liliana White Flowers		Jules Simon Pink Flowers		Prof. Hoser Blue Flowers		Andenken an Ludwig Späth Purple Flowers	
	Non-Oxidised	Oxidised	Non-Oxidised	Oxidised	Non-Oxidised	Oxidised	Non-Oxidised	Oxidised
Acetic	8.3 ± 0.21 ^d^	22.1 ± 0.6 ^b^	10.1 ± 0.3 ^c^	1.4 ± 0.02 ^f^	nd	nd	32.2 ± 0.5 a	6.5 ± 0.1 ^e^
Citric	1.3 ± 0.03 ^c^	nd	nd	1.9 ± 0.01 ^b^	4.3 ± 0.01 ^a^	nd	nd	nd
Fumaric	nd	nd	nd	nd	nd	nd	0.3 ± 0.01 ^a^	nd
Lactic	nd	nd	9.0 ± 0.2 ^c^	nd	nd	nd	15.7 ± 0.3 ^b^	182.4 ± 3 ^a^
Malic	0.7 ± 0.02 ^d^	nd	1.2 ± 0.03 ^d^	nd	nd	13.4 ± 0.2 ^a^	8.5 ± 0.1 ^c^	113.2 ± 2 ^a^
Malonic	3.2 ± 0.08 ^c^	4.8 ± 0.1 ^c^	14.9 ± 0.4 ^b^	2.8 ± 0.02 ^c^	nd	12.9 ± 0.2 ^b^	12.0 ± 0.2 ^b^	191.5 ± 3 ^a^
Oxalic	1.1 ± 0.03 ^b^	0.3 ± 0.01 ^e^	1.2 ± 0.03 ^a^	0.2 ± 0.01 ^f^	0.2 ± 0.00 ^g^	nd	0.4 ± 0.01 ^d^	0.7 ± 0.01 ^c^
Quinic	nd	9.2 ± 0.3 ^c^	2.8 ± 0.07 ^d^	nd	1.9 ± 0.01 ^e^	nd	25.3 ± 0.4 ^b^	27.4 ± 0.5 ^a^
Succinic	10.6 ± 0.3 ^d^	110.2 ± 3 ^a^	68.0 ± 1.8 ^b^	nd	nd	nd	28.3 ± 0.5 ^c^	28.3 ± 0.5 ^c^
Sum	25.2 ± 0.6 ^e^	146.5 ± 4.2 ^b^	107.2 ± 2.8 ^d^	6.4 ± 0.03 ^f^	6.4 ± 0.01 ^f^	26.4 ± 0.3 ^e^	122.6 ± 2 ^c^	550.1 ± 9.1 ^a^

*n* = 3; identical superscripts (a–f) denote non-significant differences between means in rows according to the post hoc Tukey’s HSD test; nd—not detected.

**Table 5 molecules-28-05159-t005:** Validation data.

Compound	Retention Time [min.]	Calibration Curve	R^2^	Recovery [%]
Oxalic	1.007	y = 267553x + 5542.8	0.9962	87
Quinic	1.106	y = 70546x−836.3	0.9816	95
Malonic	1.314	y = 46849x + 221.41	0.9853	97
Lactic	1.462	y = 1669.4x + 1.4949	0.9929	90
Citric	1.579	y = 70345x − 14764	0.9951	90
Acetic	1.605	y = 36463x + 17143	0.9829	95
Malic	1.895	y = 92480x − 2263.2	0.9911	92
Succinic	1.99	y = 18434x − 687.09	0.9891	100
Fumaric	2.114	y =7878614x + 480.24	0.9895	95
Gallic acid	2.28	y = 5173.6x + 52.429	0.9984	94
Protocatechuic acid	4.02	y = 4544.8x − 6.8547	0.9997	90
2,5-DHBA	7.21	y = 1053.5x + 35.048	0.9926	92
4-HBA	7.68	y = 4827.5x + 91.143	0.9983	96
Vanillic acid	9.12	y = 5224.6x + 88	0.9988	85
Catechin	9,41	y = 6086.4x − 124.43	0.9982	89
Caffeic acid	10.04	y = 9534.5x + 21.157	0.9947	91
Syringic acid	10.62	y = 7944.6x + 15.566	0.9981	94
*p*-Coumaric acid	12.18	y = 14480x − 152.61	0.9981	89
Chlorogenic acid	14.04	y = 3534x − 6.4819	0.9991	92
Ferulic acid	14.24	y = 12317x − 81.88	0.9993	91
Sinapic acid	15.19	y = 11260x − 187.39	0.9952	94
Rutin	15.52	y = 3007x − 42.036	0.9964	93
*t*-Cinnamic acid	21.41	y = 17917x + 144.58	0.9962	97
Quercetin	22.02	y = 1814.2x + 16.337	0.9986	97
Luteolin	22.82	y = 6222.2x − 62.422	0.9982	96
Apigenin	25.67	y = 9411.8x − 25.855	0.9958	93
Kaempferol	26.14	y = 4681.4x + 20.916	0.9989	87

## Data Availability

The data that support the findings of this study are available from the corresponding author upon reasonable request.

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
