# Peer review of "Phenolic Compounds and Organic Acid Composition of Syringa vulgaris L. Flowers and Infusions"

_molecules, 2023, doi:10.3390/molecules28135159_

Round 1

Reviewer 1 Report

The aim of the manuscript molecules-2449327 is to evaluate the constituents and in vitro antioxidant activity of alcoholic and aqueous extracts prepared from flowers of four Syringa vulgaris cultivars. The authors aimed to assess the influencing effect of the color of the flowers (blue, pink, purple, white) as well as that of the sample processing method (with or without enzymatic oxidation). Composition of the extracts was evaluated by two colorimetric assays (total polyphenol and flavonoid contents) and quantifying twenty-six constituents using an UHPLC-DAD method. In vitro radical scavenging activity of the extracts was surveyed by the DPPH method.

Data on the composition of antioxidant phenolics in lilac flowers would be of great interest, however, there are several major areas of concern in the manuscript,

Main remarks

Why did the authors evaluate extracts of ‘oxidised’ lilac flowers? Are infusions of ‘oxidised’ lilac flowers consumed traditionally? The authors write that ‘Flowers are crystalized in salads with egg whites and sugar [6].’ (line 25).

The authors also write that ‘Yigit et al. [2] found that herbal tea is made from lilac flowers after a special preparation. The process is similar to the production of black tea.’ (lines 25-26). However, little similarity can be found between the methods of YiÄŸit et al. (reference 2) (drying at ambient temperature or at 60 °C) and the production of black tea (withering, rolling, oxidizing and drying).

What was the purpose of oxidizing lilac flowers? In case of back tea, catechins are oxidized to form theaflavins and thearubigins that characterize the color of black tea.

The DPPH method as a single assay is not adequate in itself to precisely and quantitatively detect actions of a putative antioxidant. In addition to the DPPH method, further assays should have been employed to assess the in vitro radical scavenging activity of lilac extracts.

Why did the authors choose the very compounds listed in Tables 1-2 and Tables 3-4 as the markers representing the quality of S. vulgaris extracts? Are these the major constituents? UHPLC-DAD chromatograms characteristic of the evaluated extracts should have been included in the manuscript.

There are further abundant constituents in lilac flowers (i.e. anthocyanins) (Yiğit et al., reference 2; Deevaа et al., 2022, https://doi.org/10.1134/S1021443722020042). The total anthocyanin content of the extracts as well as quantities of individual anthocyanin compounds should have also been evaluated.

Including anthocyanin constituents should have been particularly important because of their potent antioxidant activity. According to Rice-Evans et al. (1996), delphinidin, cyanidin, and their glycosides (characteristic of lilac extracts) show Trolox Equivalent Antioxidant Activities comparable to those of quercetin and epigallocatechin (Rice-Evans et al., 1996, https://doi.org/10.1016/0891-5849(95)02227-9).

The authors refer to their previous work (reference 39) as the source of the UHPLC method applied for the quantitation of the selected compounds. However, this previous method does not include quantitation of kaempferol, apigenin, luteolin, p-coumaric acid, ferulic acid, lactic acid, and oxalic acid that have indeed been quantified in this work.

Validation data of the quantitative UHPLC analysis should have been included in the manuscript, since they are not presented in the previous work of the authors (reference 39).

Further comments

The authors refer to lilac extracts as ‘methanolic extracts’ throughout the manuscript. However, in the ‘Materials and Methods’ section they write that flower samples were extracted using 80% ethanol (line 224).

What was the final concentration of the analyzed extracts? 

Radical scavenging activity of the extracts cannot be interpreted without the knowledge of the concentration of the analyzed extracts.

Standard deviation values should have been given and shown where applicable, e.g. Fig. 1-3., Table 1-4.

Supplier companies and their locations are missing throughout the ‘Materials and Methods’ section (e.g. Folin–Ciocalteu’s phenol reagent, HPLC eluents, etc.).

The manuscript should be linguistically and stylistically proofread. The clarity of the manuscript should also be improved (e.g. lines 111-112 ‘a significant increase in the sum of determined acids was found in the S. vulgaris flowers samples after oxidation regarding the samples AFTER THE NON OXIDISED’; line 255 ’SODIUM NITRATE (III) (NaNO2)). 

Author Response

Thank you for  comments and suggestions. Each of them helped to improve the manuscript. We corrected the manuscript according to the suggestions.

Reviewer 2 Report

1- As per author "The highest content of most phenolic compounds in extracts was confirmed for purple non oxidised flowers, while in infusions for white non oxidised flowers" I suggest to include the statistical value inside the abstract.

2- Please provide humidity details on storage conditions. 

3-Please provide specification of storage area.

4-Please provide drying conditions. Drying specifications is missing.

5-Please provide extraction yield.

6-Please provide flow diagrams of extraction procedure.

7-Missing relavent citation of experimentation procedure adopted such as Determination of Total Phenolic Content, Determination of the Total Flavonoid Content, and DPPH assay. Please do mentioned if modification of experiment from literature.

8-Please include heading of instrumentation inside the methodology.

9-Please provide study stength and study limitation as a heading and explain.

10- There is no practical value mentioned inside the conclusion. Please write a separate heading and explain.

11- I am not convinced with the conclusion written. Discuss the implication of study. rewrite with more significance of your findings and future directions. 

quality of language is okay.

Author Response

(The authors gave the same response as above.)

Reviewer 3 Report

The manuscript “Phenolic Compounds and Organic Acids Composition of Syringa vulgaris L. Extracts and Infusions” deals with the determination of phenolic compounds and organic acids from oxidized and non-oxidized flowers. The manuscript presents interesting results, it is well-organized and well-written. I believe that this article is worth publication but needs revisions. I have some in the following comments.

1) Abstract should bring some relevant numerical values of the results to a better elucidation of the work.

2) In Section 4.2. the authors should mention the brand and country of fabrication of the materials used (e.g., liquid nitrogen, ethanol, ultrasound apparatus, shaker, evaporator).

3) Did the authors follow any methodology for the phenolic compounds and organic acids determination (Section 4.3)?

4) For Sections 4.4 to 4.6 it is necessary to say what methodology was followed to the analysis. Reagents should have their brand and country of fabrication mentioned for a better understanding of the readers.

5) Section 4.6. should mention the degree of purity of each standard used.

6) In Section 4.7., what did the authors mean by the expression “tree replicas”? Would not that be “three replications”?

7) Conclusion should bring some suggestions for future works and challenges for the ones who want to continue this field of study.

The paper is well-written and English grammar is acceptable.

Author Response

(The authors gave the same response as above.)

Round 2

Reviewer 1 Report

The manuscript molecules-2449327-v2 has been improved, additional data have been added as requested: on the background (significance of oxidized lilac flower infusions), the methodology (ABTS radical scavenging activity), the results (standard deviation values) as well as the methods and materials applied (parameters of the UHPLC method, validation data, supplier details).

Further minor comments are included below:

Figure 3 should present UHPLC-DAD chromatograms of both non-oxidized and oxidized extracts, thus, demonstrating the characteristic differences caused by the oxidation.

UHPLC-DAD chromatograms of which extract are presented in Figure 3? This should be indicated in the figure caption.

Detection wavelength of the UHPLC-DAD chromatograms should also be indicated not only in the figure caption but also in Figure 3.

Quality (resolution) of Figure 3 should be improved.

Concentrations of the analyzed extracts should be indicated in Figures 4 and 5 (or in their respective figure captions). The authors might consider presenting IC50 data.

Heading of Figures 4 and 5 should be more precise, the expressions ‘DPPH in flowers [%]’ and ‘ABTS in flowers [%]’ are not appropriate.

Full names of compounds (e.g. 4-HBA, 2,5-DHBA) should be spelled out upon first mention.

Author Response

Thank Reviewer for another comments and suggestions. Each of them helped to improve the manuscript.

Unfortunately, members of our team (chemists) are currently abroad on a research internship and there is no possibility to access the apparatus to complete the data of figure 3.
